# Electrochemical oxidative difunctionalization of diazo compounds with two different nucleophiles

Dongfeng Yang[1,6], Zhipeng Guan[1,6], Yanan Peng[1], Shuxiang Zhu[1], Pengjie Wang[1], Zhiliang Huang[1], Hesham Alhumade[2,3], Dong Gu[1] ✉, Hong Yi[1,4] ✉ & Aiwen Lei[1,5] ✉

With the fast development of synthetic chemistry, the introduction of functional group into organic molecules has attracted increasing attention. In these reactions, the difunctionalization of unsaturated bonds, traditionally with one nucleophile and one electrophile, is a powerful strategy for the chemical synthesis. In this work, we develop a different path of electrochemical oxidative difunctionalization of diazo compounds with two different nucleophiles. Under metal-free and external oxidant-free conditions, a series of structurally diverse heteroatom-containing compounds hardly synthesized by traditional methods (such as high-value alkoxy-substituted phenylthioacetates, $\alpha$-thio, $\alpha$-amino acid derivatives as well as $\alpha$-amino, $\beta$-amino acid derivatives) are obtained in synthetically useful yields. In addition, the procedure exhibits mild reaction conditions, excellent functional-group tolerance and good efficiency on large-scale synthesis. Importantly, the protocol is also amenable to the key intermediate of bioactive molecules in a simple and practical process.

Diazo compounds are a class of charge-neutral organic compounds containing diazo groups bound to carbon atoms that can resonate in different structures[1–3]. In this regard, chemists have been devoting continuous efforts to pursuing the versatile transformations of diazo compounds for the synthesis of a variety of high-value molecules, such as pharmaceuticals and natural products[4–8]. Among them, the difunctionalization of diazo compounds has emerged as an important tool for the formation of challenging carbon-carbon and carbon-heteroatom bonds with the introduction of multiple functional groups[9–17]. Over the last decades, transition-metal (such as Rh, Cu, Fe, Pd, etc) catalyzed difunctionalization of diazo compounds with nucleophiles and electrophiles has undergone a flourishing development because of the unique reaction properties of metal carbene (Fig. 1a)[18–27]. In addition, thermolysis or photolysis induced carbene transfer reactions provide alternative routes for the difunctionalization of diazo compounds[28–32]. On the other hand, due to the electron-rich nature of the resonance structures of the diazonium compounds, the diazonium compounds can react with electrophiles through a carbene-free mechanism. And then the diazonium ion intermediate is captured by nucleophile in an intermolecular or intramolecular manner to construct target molecules (Fig. 1b)[33–37]. Despite these excellent developments, a more attractive, a green and atom-economical approach to realize the difunctionalization of diazo compounds with readily available starting materials is still highly anticipated and remains a relevant challenge. As shown in Scheme 1c, if two different, but readily available nucleophiles are employed in the difunctionalization of diazo compounds under mild conditions, it will present an interesting and different avenue to discovery and build organic molecules[38–40].

[1]The Institute for Advanced Studies (IAS) and College of Chemistry and Molecular Sciences, Wuhan University, Wuhan 430072 Hubei, P. R. China. [2]K. A. CARE Energy Research and Innovation Center, King Abdulaziz University, Jeddah 21589, Saudi Arabia. [3]Department of Chemical and Materials Engineering, Faculty of Engineering, Center of Research Excellence in Renewable Energy and Power Systems, King Abdulaziz University, Jeddah 21589, Saudi Arabia. [4]Wuhan University Shenzhen Research Institute, Wuhan University, Shenzhen 518057, P. R. China. [5]Chemical and Materials Engineering Department, Faculty of Engineering, King Abdulaziz University, Jeddah 21589, Saudi Arabia. [6]These authors contributed equally: Dongfeng Yang, Zhipeng Guan. ✉e-mail: DGu@whu.edu.cn; hong.yi@whu.edu.cn; aiwenlei@whu.edu.cn

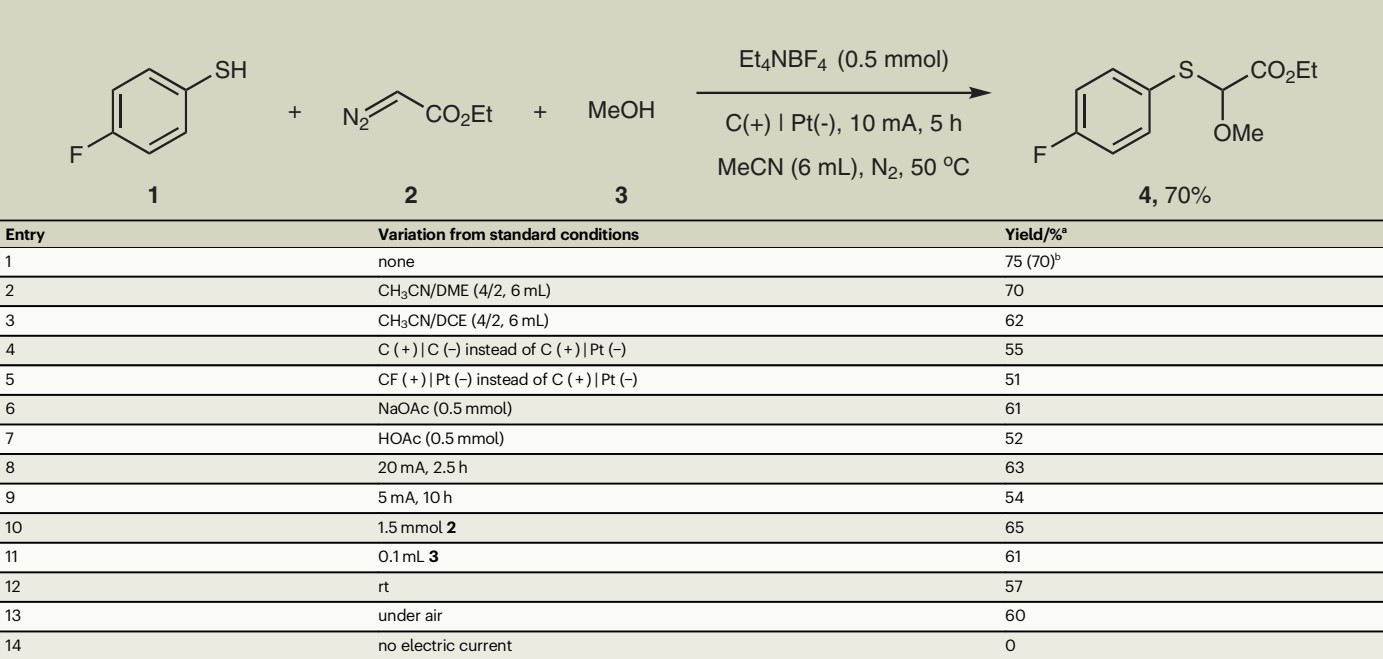

**Fig. 1 | Reaction models of diazo compounds. a** The carbene reactivity of diazo compounds. **b** Process via diazonium ion intermediates. **c** Our design: Electrochemical oxidative difunctionalization of diazo compounds.

## Table 1 | Optimization of the reaction conditions

| Entry | Variation from standard conditions | Yield/%[a] |
|---|---|---|
| 1 | none | 75 (70)[b] |
| 2 | CH₃CN/DME (4/2, 6 mL) | 70 |
| 3 | CH₃CN/DCE (4/2, 6 mL) | 62 |
| 4 | C (+) | C (−) instead of C (+) | Pt (−) | 55 |
| 5 | CF (+) | Pt (−) instead of C (+) | Pt (−) | 51 |
| 6 | NaOAc (0.5 mmol) | 61 |
| 7 | HOAc (0.5 mmol) | 52 |
| 8 | 20 mA, 2.5 h | 63 |
| 9 | 5 mA, 10 h | 54 |
| 10 | 1.5 mmol **2** | 65 |
| 11 | 0.1 mL **3** | 61 |
| 12 | rt | 57 |
| 13 | under air | 60 |
| 14 | no electric current | 0 |

Standard conditions: **1** (0.5 mmol), **2** (2.0 mmol), **3** (0.5 mL), Et₄NBF₄ (0.5 mmol), CH₃CN (6.0 mL) in an undivided cell with carbon rod as anode, platinum as cathode (1.5 × 1.5 cm²), constant current = 10 mA, 50 °C, N₂, 5 h.
[a]¹⁹F NMR yield, 3-fluorotoluene as an internal standard.
[b]Yield of isolated product.

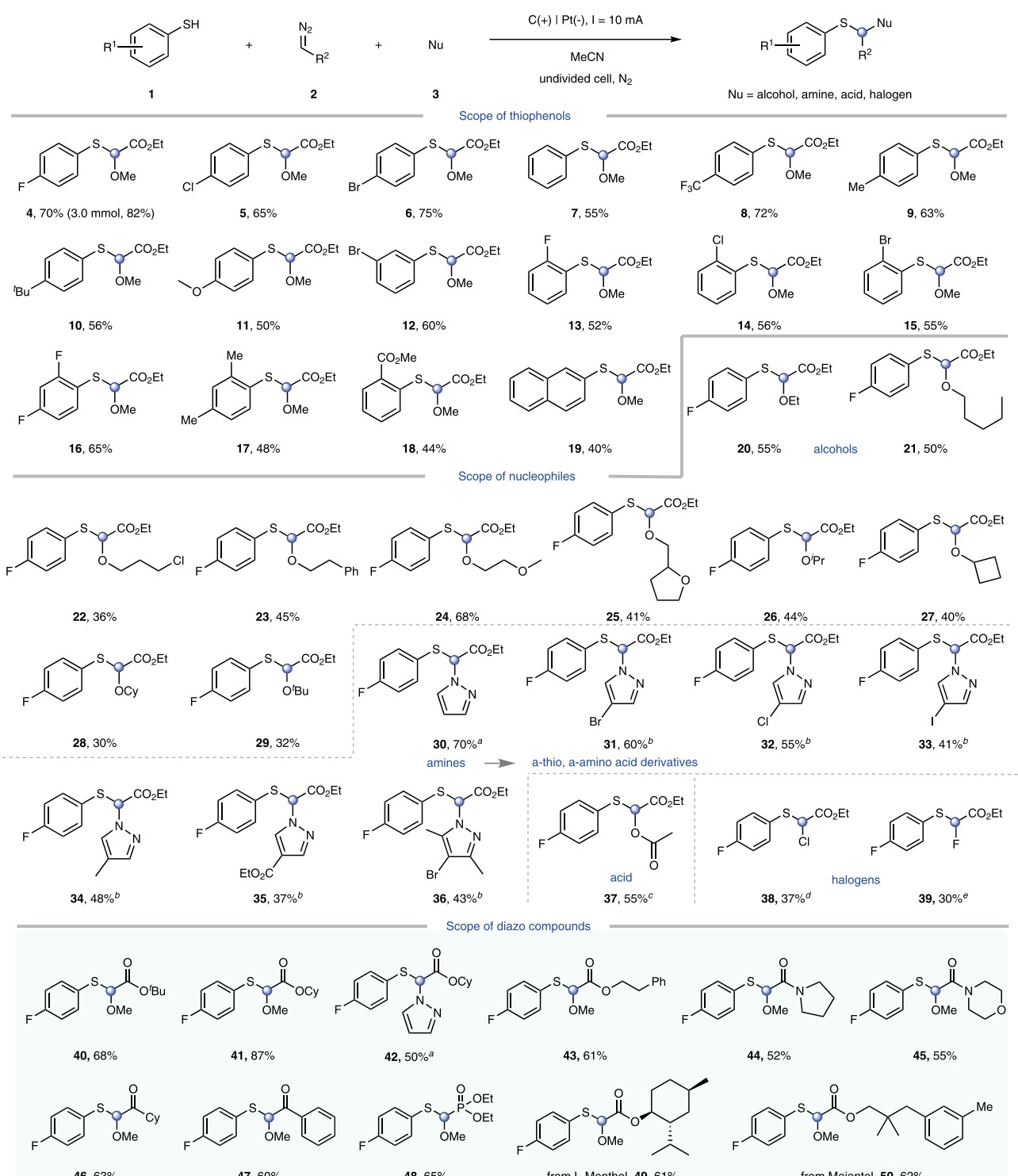

**Fig. 2 | Scope of thiophenols, nucleophiles and diazo compounds.** Standard conditions: thiolating agents (0.5 mmol), diazo compounds (2.0 mmol), alcohols (0.5 mL), Et$_4$NBF$_4$ (0.5 mmol), MeCN (6 mL), carbon rod as anode, platinum as cathode, undivided cell, constant current = 10 mA, 50 °C, N$_2$, 5 h, yield is that of the isolated product. [a]Used pyrazole (2.0 mmol) instead of alcohol. [b]Used pyrazole (2.0 mmol) instead of alcohol, constant current = 10 mA, rt, 7 h. [c]Used HOAc (0.5 mL) instead of alcohol. [d]Used LiCl (2.0 mmol), H$_2$O (50 µL) instead of alcohol, constant current = 10 mA, 50 °C, 7 h. [e]Used Et$_3$N·3HF (0.5 mL) instead of alcohol, constant current = 10 mA, 50 °C, 7 h.

In recent years, the electrochemistry-induced reactions have drawn much attention due to the high reaction efficiency, innate scalability and sustainability of the electrolytic process[41–47]. The past decade has witnessed the rapid development of difunctionalization of unsaturated bonds in organic synthesis[48–54]. Despite significant success in the area, most of these transformations are limited to the functionalization of two carbon atoms of alkenes and alkynes because of the intrinsic reaction nature[55–61]. In order to expand the application of other unsaturated bonds, we developed the electrochemical oxidative carbon-atom difunctionalization of isocyanides for the construction of multi-substituted imino sulfide ethers, isothioureas as well as α-amino amides[62–64]. Based on our previous efforts in electro-chemistry, we

**Fig. 3 | Scope of substrates of carbon radicals.** Conditions: [a]aniline (0.5 mmol), ethyl diazoacetate (0.3 mmol), HOAc (0.5 mL), $^{n}Bu_4NBF_4$ (1.0 mmol), MeCN (6 mL), carbon rod as anode, platinum as cathode, undivided cell, constant current = 10 mA, rt, $N_2$, 2 h; isolated yield. [b]aniline (0.7 mmol), ethyl diazoacetate (0.3 mmol), LiCl (0.3 mmol), $H_2O$ (50 μL), HOAc (0.5 mL), $^{n}Bu_4NBF_4$ (1.0 mmol), MeCN (6 mL), carbon rod as anode, platinum as cathode, undivided cell, constant current = 10 mA, rt, $N_2$, 2.25 h.

envision that tandem single-electron oxidation may realize the difunctionalization of diazo compounds with two different nucleophiles via electrochemical approach: a radical generated from single-electron oxidation of a nucleophile on anode could be captured by diazo compound to form a carbon radical intermediate. The fast second single-electron oxidation of the intermediate suppressed the decomposition of the radical, and further was attacked by another nucleophile to generate the desired difunctionalization molecules (Fig. 1c).

Herein, we develop a different path of electrochemical oxidative difunctionalization of diazo compounds with two different nucleophiles. A series of structurally diverse heteroatom-containing compounds hardly synthesized by traditional methods (such as high-value alkoxy-substituted phenylthioacetates, α-thio, α-amino acid derivatives as well as α-amino, β-amino acid derivatives) are obtained in synthetically useful yields.

## Results and discussion
### Condition optimization for electrochemical oxidative difunctionalization of diazo compounds

We started to investigate the electrochemical oxidative difunctionalization of ethyl diazoacetate (**2**) with 4-fluorothiophenol (**1**) as the radical source and methanol (**3**) as the nucleophile. Although there are several challenges in this transformation: (1) competitive difunctionalization with same nucleophiles; (2) competitive monofunctionalization instead of difunctionalization owing to properties of intermediate, to our delight, utilizing carbon rod as anode, and platinum plate as cathode under 10 mA constant current in undivided cell, the desired ethyl 2-((4-fluorophenyl)thio)−2-methoxyacetate (**4**) was obtained selectively with 70% isolated yield (Table 1, entry 1). No better reaction efficiency was observed in this electrolysis by using other solvent system, such as $CH_3CN/DME$ (4:2) and $CH_3CN/DCE$ (4:2) (Table 1, entries 2 and 3). The use of carbon rod as cathode or graphite felt as anode resulted in a decrease in yield (Table 1, entries 4 and 5). The addition of NaOAc or HOAc to the reaction showed a negative effect on product formation (Table 1, entries 6 and 7). The increase or decrease of the reaction current slightly reduced the product yield (Table 1, entries 8 and 9). While a lower ethyl diazoacetate of 1.5 mmol or a lower amount of methanol showed a decrease in the amount of the

target compound (Table 1, entries 10 and 11). Further research showed that the reaction was feasible at ambient temperature (Table 1, entry 12). The reaction proceeds smoothly in an air atmosphere, which highlights the utility of this protocol (Table 1, entry 13). The electricity was critical for the transformation (Table 1, entry 14).

### Scope of substrates

With the confirmation of optimized conditions, the scope of thiophenol was explored under standard conditions (Fig. 2). Firstly, the electronic effect of aryl-substitution was investigated with various *para*-substituted thiophenols. It was found that halogenated thiophenols (such as aryl fluoride **4**, aryl chloride **5** and aryl bromide **6**) were tolerated, which could induce subsequent catalyst conversion of the products under the corresponding conditions. Gratifyingly, product **4** was obtained with a higher yield (82%) in the scale-up of this electrosynthesis. The unsubstituted thiophenol also readily adopted in this protocol (**7**). Moreover, thiophenols with electron-withdrawing groups ($CF_3$, $CO_2Me$) or electron-donating groups (Me, $^{t}Bu$, and OMe) were able to afford the corresponding products in satisfactory yields (**8–11**, **18**). This reaction could proceed effectively with thiophenols containing *meta*-group and sterically impeding *ortho*- halogen/methyl/ester functionality, which delivered a variety of target molecules in moderate yields (**12–18**). Naphthalene-2-thiol was also compatible with this transformation (**19**). With respect to nucleophile coupling partner, we selected 4-fluorobenzenethiol **1** to interrogate the performance of a range of alcohols, azoles and acids under the electrochemical conditions. The catalytic efficiency of the reaction showed little effect on the chain length of primary alcohols (**20–21**). 3-chloropropan-1-ol was tolerated to deliver the desired product albeit in a modest yield (**22**). The reaction was performed with other fatty alcohols, such as phenethyl alcohol (**23**), 2-methoxyethanol (**24**) and tetrahydrofurfuryl alcohol (**25**). Secondary alcohols and tertiary alcohol could be used successfully in the electrolysis, representing a practical route to construct C-S and C-O bonds build on the same carbon atom (**26–29**). Importantly, a series of α-thio, α-amino acid derivatives were obtained successfully with amine substrates obtaining various groups, such as halogen, methyl and ester, showcasing the versatility of this reaction platform (**30–36**). Besides alcohols and azoles, acetic acid was also a

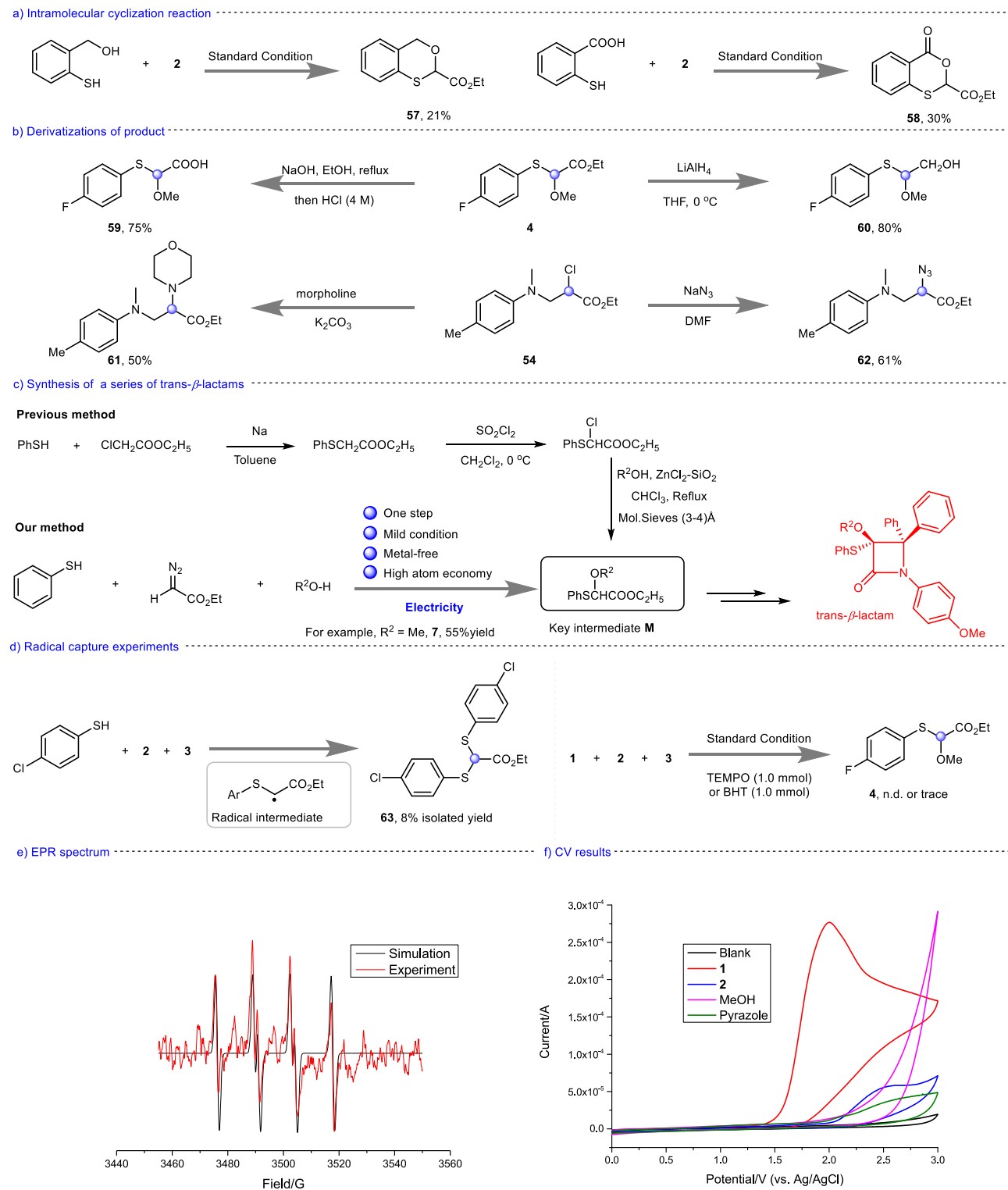

**Fig. 4 | The application of the method and the mechanistic experiments. a** Intramolecular cyclization reaction. **b** Derivatizations of product. **c** Synthesis of a series of trans-$\beta$-lactams. **d** Radical capture experiments. **e** EPR spectrum. **f** CV results.

suitable nucleophile for this protocol (**37**). Finally, fluorinated or chlorinated products could be accessed by utilizing lithium chloride (LiCl) or triethylamine trihydrofluoride (Et$_3$N·3HF) as halogen sources (**38–39**).

We next investigated the generality of diazo compounds under the established conditions. *tert*-Butyl 2-diazoacetate, cyclohexyl 2-diazoacetate and phenethyl 2-diazoacetate could be smoothly converted into the corresponding difunctionalization products in moderate to good yields (**40–43**). Besides diazoacetates, other diazo compounds were used in the transformation. To our delight, diazoamide, diazoalkylketone, diazoarylketone and diazophosphonate were also coupling partners, which could react smoothly with thiophenol and methanol, providing a series of functionalized products (**44–48**). Pleasingly, segments of natural products (such as menthol and majantol) were amenable to this protocol, which further highlighted the potential value of this methodology (**49–50**).

To further expand the scope of the difunctionalization of diazo, we envisaged the use of carbon radicals instead of above sulfur radicals (Fig. 3). Unexpectedly, we found that *N*, *N*-dimethylaniline was compatible with the difunctionalization of diazo. After a series of optimization of conditions, the target product ethyl−2-acetoxy-3-(methyl(*p*-tolyl)amino)propanoate (**51**) was obtained in 69% isolated yield by using carbon rod as an anode, platinum plate as a cathode at room temperature and a constant current of 10 mA in MeCN system. The similar *β*-amino acid derivatives (**52**–**53**) were isolated via electrochemical oxidative C(sp³)-H activation. Using LiCl as a nucleophile, this transformation could proceed smoothly to construct C-C and C-Cl bonds build on the same carbon atom simultaneously (**54**–**56**). hydride in tetrahydrofuran. The carboxy and oxhydryl provide a functional group handle for further derivatization. In addition, the active chlorine moiety was then subjected to transformations of other high-value functional groups. For example, *α*-chloro, *β*-amino acid ester was treated with morpholine to obtain *α*-amino, *β*-amino acid derivative in the presence of K₂CO₃. When **54** was then treated with NaN₃, the expected *α*-azide, *β*-amino acid derivative was obtained in 61% yield. Importantly, *β*-lactam compounds are widely used in antibiotics, inhibitors, and drug structures. And alkoxy-substituted phenylthioacetates are important precursors (**M**) for their synthesis. It had been reported in relevant literature that the preparation of alkoxy substituted phenylthioacetate (**M**) required the use of hazardous and sensitive SO₂Cl₂ and metal sodium as well as multi-step synthesis. Herein, we have developed a green, fast, scalable, and metal-free method to synthesize the key intermediate **M** with high atom efficiency and the only release of hydrogen and nitrogen (Fig. 4c).

## Synthetic applications

We have performed the intramolecular cyclization reaction. Using 2-mercaptobenzyl alcohol as a substrate, the target product (**57**) was isolated with 21% yield. Using 2-mercaptobenzoic acid as a substrate, the target product (**58**) was isolated with 30% yield (Fig. 4a). The versatility of alkoxy-substituted phenylthioacetate was further explored (Fig. 4b). For instance, the carboxylation process of **4** led to the formation of 2-((4-fluorophenyl)thio)−2-methoxyacetic acid (**59**) in 75% yield by using NaOH as base. **4** could also be reduced to 2-((4-fluorophenyl)thio)-2-methoxyethan-1-ol (**60**) with lithium aluminum

## Mechanistic studies

To more deeply study the mechanism, we carried out a series of control experiments. Under the standard conditions, ethyl 2,2-bis((4-chlorophenyl)thio)acetate **63** was isolated in 8% yield, which indicated the radical intermediate was involved in the transformation (Fig. 4d). The target product **4** was undetected when 2.0 equivalent of TEMPO was added to the reaction system. Adding 2.0 equivalent BHT into the reaction resulted in trace amounts of product **4**. These experiments indicated the electrolysis may undergo a radical process (Scheme 4d). In addition, we also carried out electron paramagnetic resonance (EPR) experiments (Fig. 4e). A S-centered radical was rapidly trapped by 5,5-dimethyl-1-pyrroline *N*-oxide (DMPO) to form a relatively stable radical (g = 2.0070, AN = 13.40 G, AH = 14.88 G). The cyclic voltammetry (CV) experiments were carried out to further understand the details of the oxidation process (Fig. 4f). The results showed that the 4-fluorothiophenol (**1**) has an oxidation potential of 1.98 V vs. Ag/AgCl, while the oxidation potential of ethyl diazoacetate (**2**) for Ag/AgCl was over 2.60 V. These results indicated that 4-fluorothiophenol was preferentially oxidized under optimized conditions.

Based on the above mechanism studies and related literature reports, we proposed a possible mechanism for the electrochemical oxidative difunctionalization of diazo, as shown in Fig. 5. Firstly, thiophenol lost a proton to product sulfur anion and further was oxidized by a single-electron-transfer to obtain a sulfur radical at the anode. The

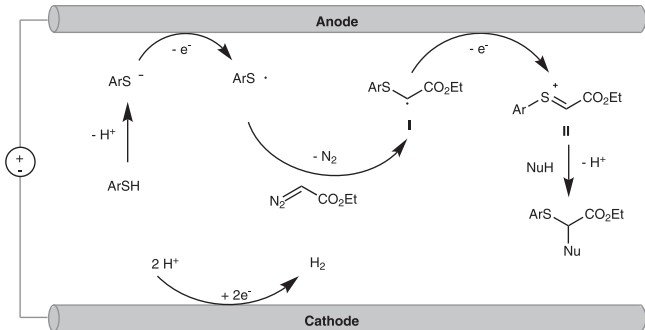

**Fig. 5 | Proposed reaction mechanism.** Electrochemical oxidative difunctionalization of diazo.

sulfur radical could be captured by diazo compound to generate a carbon radical **I** with the release of nitrogen. Subsequently, radical **I** underwent rapid single-electron oxidation to generate cation **II**. Finally, the intermediate **II** was subjected to intermolecular nucleophilic attack by the nucleophile to form the desired product. The proton was reduced on the cathode to release hydrogen gas.

In conclusion, an interesting electrochemical oxidative difunctionalization of diazo compounds has been developed. Cheap and commercially available radical sources (such as S and C) and nucleophiles (such as alcohol, amine, acid and halogen anion) could be employed to perform the reaction. This simple procedure avoids the use of transition metals and external oxidants, the pre-activation of substrates (such as electrophiles) as well as harsh conditions. Excellent functional group compatibility, good large-scale synthesis efficiency and diverse synthetic application further highlight the strengths of this transformation.

## Methods

In an oven-dried undivided three-necked bottle (10 mL) equipped with a stir bar, Et₄NBF₄ (0.5 mmol) was added. The bottle was equipped with carbon rod as the anode and platinum plate (1.5 cm × 1.5 cm × 1 mm) as the cathode. Under nitrogen atmosphere, thiophenol (0.5 mmol), diazo compounds (2.0 mmol), alcohols (0.5 mL) and MeCN (6.0 mL) were injected respectively into the tube via syringes. The reaction mixture was stirred and electrolyzed at a constant current of 10 mA for 5 h at 50 °C. The reaction mixture was washed with water and extracted with EA (10 mL × 3). The organic layers were combined, dried over Na₂SO₄ and concentrated in vacuum. The pure product was purified by silica gel flash column chromatography.

## Data availability

The authors declare that the data supporting the findings of this study are available within the paper and its supplementary information files or from the authors upon request.

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

## Acknowledgements

This work was supported by the National Key R&D Programof China (No. 2022YFA1505100 and 2021YFA1500104), a project fundedby the China Postdoctoral Science Foundation (2021M702516), the National Natural Science Foundationof China (22031008), Guangdong Basic and Applied Basic Research Foundation (2023A1515012260) and the Science Foundation of Wuhan (2020010601012192), the Fundamental Research Funds for the Central Universities (2042022rc0030).

## Author contributions

D.Y. and Z.G. conceived the work. D.Y., Z.G., P.W., H.Y., and A.L., designed the experiments and analyzed the data. D.Y., Y.P., and S.Z. performed the experiments. Z.G., D.Y., Z.H., H.A., D.G., H.Y., and A.L. described the paper.

## Competing interests

The authors declare no competing interests.
