## [Peer Review File · Nature Communications]

REVIEWER COMMENTS

Reviewer #1 (Remarks to the Author):

The presented work shows an interesting difunctionalization of diazocompounds utilizing electrooxidative conditions. Two different nucleophiles are added, expanding the scope of the previously known reactions, and realizing this transformation under relatively mild conditions. This reviewer thinks that the mechanism has novelty and is evidenced with various experiments, and the reaction has tolerance to several functional groups, as well as a broad scope not only using thiols as radical precursors, but also aniline derivatives. The SI is complete, and products isolated have enough purity for publication. For these reasons, this reviewer considers that this manuscript is suitable for its publication in Nature Communications. There are some questions that arise:

1. Did the authors try to use two platinum electrodes. Using a carbon rod electrode is much desirable, but since the cathode could not be a carbon electrode, maybe changing the anode to a platinum electrode could improve the yield.
2. Is there an explainable reason for the diminished yield running the reaction open to air? This scenario would be much more desirable than using N₂ in terms of greenness and easy set up. Are there undesired byproducts formed that stall the reaction?
3. In the synthesis of 22, that was obtained in modest yield, did the authors isolate a product derived from the cyclization at the C bearing the halogen? This could be a competing reaction (non-electrochemical) due to the presence of an electrophile in the media.
4. In page 7, line162, please replace 59 by 61 (product isolated in 8% yield).

Reviewer #2 (Remarks to the Author):

This manuscript reports an electrochemical oxidative difunctionalization of diazo compounds. Different from the existing methods using transition metal catalysis or photocatalysis, this work develops a different path of electrochemical oxidative difunctionalization of diazo compounds with two different nucleophiles. As we have seen, different radical sources (thiophenols and amines) and structurally diverse nucleophiles (alcohols, acids, halogens, pyrazole) can participate in this mild

reaction. A reliable mechanism is proposed after some control experiments and cyclic voltammetry experiments were conducted. The results described by the authors are informative and interesting to Nature Communications readers, particularly those working on the electroorganic chemistry. Based on the content and quality of this paper, I think this work is suitable for Nature Communications however some issues should be addressed before the accepting this manuscript for publication.

- 1) The author showed a series of thiophenol substrates. I want to know how mercaptan reacts. If not, what are the possible reasons.
- 2) If the thiophenol substrate contains a nucleophilic group, whether intramolecular cyclization reaction can occur? For example, using 2-Mercaptophenol or 2-Mercaptobenzyl alcohol as substrate.
- 3) Finally, I want to know whether the author has tried the asymmetric synthesis of this reaction?
- 4) Repeatability is a key factor for electrochemistry. It is suggested that the author put some photo of electrolysis device in the supporting information.

Reviewer #3 (Remarks to the Author):

This manuscript reports an oxidative difunctionalization of diazo compounds with two different nucleophiles. This reaction provided a metal and oxidant free electrochemical method for the synthesis of structurally diverse heteroatom-containing compounds. Compared with previous reports on the realization of electrochemical oxidative carbon-atom difunctionalization of isocyanides and metal catalyzed carbene difunctionalization with nucleophile and electrophile, this method features a novel approach tolerated a broad scope of nucleophiles (such as alcohol, amine, acid and halogen anion) and commercially available radical sources (such as S and C). However, limitation of this work is the potential application of this work, which forms simple molecules that will not be able to attract immediate interest to a broad readership, and this would be the major concern before ready for publication. In terms of novelty of the reaction, this reviewer suggest acceptance after address the following issues.

Other observations:

- 1) In Scheme 4c, product number "61" was "59" in the corresponding discussion part.

- 2) For the control experiments with TEMPO and BHT, what happens for the reaction? Is there any byproduct or decomposition for the materials?
- 3) The reason why cation II prefers to addition with nucleophile rather than thiophenol should be given.
- 4) The cyclic voltammetry (CV) experiments for the nucleophiles should also provide.
- 5) How about other type of diazo compounds? Such as phenyldiazoacetate.
- 6) In the reaction with N, N-dimethylaniline, could this reagent be oxidized to the iminium ion species, followed by addition with diazo group?
- 7) What happens on the Cathode in the reaction with "LiX"?

Reviewer #1 (Remarks to the Author):

The presented work shows an interesting difunctionalization of diazocompounds utilizing electrooxidative conditions. Two different nucleophiles are added, expanding the scope of the previously known reactions, and realizing this transformation under relatively mild conditions. This reviewer thinks that the mechanism has novelty and is evidenced with various experiments, and the reaction has tolerance to several functional groups, as well as a broad scope not only using thiols as radical precursors, but also aniline derivatives. The SI is complete, and products isolated have enough purity for publication. For these reasons, this reviewer considers that this manuscript is suitable for its publication in Nature Communications.

Responses:

Thank you for these favorable comments on our work.

There are some questions that arise:

1. Did the authors try to use two platinum electrodes. Using a carbon rod electrode is much desirable, but since the cathode could not be a carbon electrode, maybe changing the anode to a platinum electrode could improve the yield.

Responses:

Thank you for the suggestion. We have performed the experiment with platinum electrode as anode. However, the lower yield (54%) was obtained, and the byproduct was detected by GC-MS (Figure R1), as shown in Scheme R1.

Scheme R1. The experiment with platinum electrode as anode.

Figure R1. The detection of byproduct.

2. Is there an explainable reason for the diminished yield running the reaction open to air? This scenario would be much more desirable than using N₂ in terms of greenness and easy set up. Are there undesired byproducts formed that stall the reaction?

Responses:

Thank you for the question. The lower yield (60%) was obtained when we performed the experiment under air conditions, as shown in Scheme R2. And this byproduct was considered as ethyl 2-((4-fluorophenyl)thio)-2-hydroxyacetate by GC-MS (Figure R2) since the radical intermediate may be captured by O₂ under air conditions.

Scheme R2. The experiment under air condition.

Figure R2. The detection of byproduct.

3. In the synthesis of **22**, that was obtained in modest yield, did the authors isolate a product derived from the cyclization at the C bearing the halogen? This could be a competing reaction (non-electrochemical) due to the presence of an electrophile in the media.

Responses:

Thank you for the question. In the synthesis of **22**, we did not isolate a product derived from the cyclization at the C bearing the halogen. However, the byproduct was detected by GC-MS (Figure R3), as shown in Scheme R3.

Scheme R3. The synthesis of **22**.

流路号: 1 保留时间: 21.355 (扫描数: 3072)
 质量峰: 355
 原始模式: 单个 21.355 (3072) 基峰: 139.00 (2893503)
 背景模式: 无 组 1 - 事件 1 Scan

Figure R3. The detection of byproduct.

4. In page 7, line 162, please replace 59 by 61 (product isolated in 8% yield).

Responses:

Thank you for pointing out typo. We have revised it in the revised manuscript.

Reviewer #2 (Remarks to the Author):

This manuscript reports an electrochemical oxidative difunctionalization of diazo compounds. Different from the existing methods using transition metal catalysis or photocatalysis, this work develops a different path of electrochemical oxidative difunctionalization of diazo compounds with two different nucleophiles. As we have seen, different radical sources (thiophenols and amines) and structurally diverse nucleophiles (alcohols, acids, halogens, pyrazole) can participate in this mild reaction. A reliable mechanism is proposed after some control experiments and cyclic voltammetry experiments were conducted. The results described by the authors are informative and interesting to Nature Communications readers, particularly those

working on the electroorganic chemistry. Based on the content and quality of this paper, I think this work is suitable for Nature Communications however some issues should be addressed before the accepting this manuscript for publication.

Responses:

Thank you for these favorable comments on our work.

1) The author showed a series of thiophenol substrates. I want to know how mercaptan reacts. If not, what are the possible reasons.

Responses:

Thank you for the question. Using butane-1-thiol as radical source, the target product was not detected and the byproduct was detected by GC-MS (Figure R4). The carbon radical is more easily captured by sulfur radical rather than oxidized to produce carbon cation, as shown in Scheme R4.

Scheme R4. The experiment with butane-1-thiol as radical source.

流路号:1 保留时间:18.445(扫描数:2490)
 质量峰:354
 原始模式:单个 18.445(2490) 基峰:191.05(507756)
 背景模式:无 组 1 - 事件 1 Scan

Figure R4. The detection of byproduct.

2) If the thiophenol substrate contains a nucleophilic group, whether intramolecular cyclization reaction can occur? For example, using 2-Mercaptophenol or 2-Mercaptobenzyl alcohol as substrate.

Responses:

Thank you for the suggestion. We have performed the intramolecular cyclization reaction, as shown in Scheme R5. Using 2-mercaptophenola as a substrate, the target product was detected by GC-MS (Figure R5); Using 2-mercaptobenzyl alcohol as a substrate, the target product was isolated with 21% yield. In addition, using 2-mercaptobenzoic acid as a substrate, the target product was isolated with 30% yield. We have added these experiment results into the revised manuscript.

Scheme R5. These experiments with 2-mercaptophenola, 2-mercaptobenzyl alcohol and 2-mercaptobenzoic acid as substrates.

Figure R5. The detection of target product.

3) Finally, I want to know whether the author has tried the asymmetric synthesis of this reaction?

Responses:

Thank you for the comment. Although we have tried some chiral phosphoric acid, such as (*S*)-3,3'-Bis(2,4,6-triisopropylphenyl)-1,1'-binaphthyl-2,2'-diyl hydrogenphosphate,

the result was not satisfactory.

(S)-3,3'-Bis(2,4,6-triisopropylphenyl)-1,1'-binaphthyl-2,2'-diyl hydrogenphosphate

Scheme R6. The structure of chiral phosphoric acid.

4) Repeatability is a key factor for electrochemistry. It is suggested that the author put some photo of electrolysis device in the supporting information.

Responses:

Thank you for the suggestion. We took some photos of electrolysis device and put these photos in the revised SI.

Figure R6. Some photos of electrolysis device.

Reviewer #3 (Remarks to the Author):

This manuscript reports an oxidative difunctionalization of diazo compounds with two different nucleophiles. This reaction provided a metal and oxidant free electrochemical method for the synthesis of structurally diverse heteroatom-containing compounds. Compared with previous reports on the realization of electrochemical oxidative carbon-atom difunctionalization of isocyanides and metal catalyzed carbene difunctionalization with nucleophile and electrophile, this method features a novel approach tolerated a broad scope of nucleophiles (such as alcohol, amine, acid and halogen anion) and commercially available radical sources (such as S and C). However, limitation of this work is the potential application of this work, which forms simple molecules that will not be able to attract immediate interest to a broad readership, and this would be the major concern before ready for publication. In terms of novelty of the reaction, this reviewer suggest acceptance after address the following issues.

Other observations:

1) In Scheme 4c, product number “61” was “59” in the corresponding discussion part.

Responses:

Thanks for pointing out the typo. We have revised it in the revised manuscript.

2) For the control experiments with TEMPO and BHT, what happen for the reaction?

Is there any byproduct or decomposition for the materials?

Responses:

Thanks very much for your comments. Adding the TEMPO, sulfur radicals generated the corresponding disulfides as shown in Figure R7; Adding the BHT, the detection of GC-MS indicated that some sulfur radicals were captured by BHT and byproduct ethyl 2-((3-fluorophenyl)thio)-2-((4-fluorophenyl)thio)acetate was generated, as shown in Figure R8. Most of the ethyl diazoacetate was detected by TLC.

流路号:1 保留时间:18.590(扫描数:2519)
质量峰:337
原始模式:单个 18.590(2519) 基峰:127.00(7382302)
背景模式:无 组 1 - 事件 1 Scan

Figure R7. The detection of disulfides.

流路号:1 保留时间:21.790(扫描数:3159)
 质量峰:354
 原始模式:单个 21.790(3159) 基峰:219.10(4085652)
 背景模式:无 组 1 - 事件 1 Scan

Figure R8. The detection of **BHT-S**.

3) The reason why cation II prefers to addition with nucleophile rather than thiophenol should be given.

Responses:

Thank you for the comment. We believe that the oxidation process occurred nearing anode surface or double layer, as shown in Scheme 7. Firstly, the amount of ArSH is different from nucleophile. Secondly, during the reaction, ArSH was oxidized to generate ArS[·], thus, the amount of ArSH or ArS[·] sharply decreased. Overall, when the cation II was formed, the amount of ArSH or ArS[·] was small, but MeOH (nucleophile) was abundant in the system.

Scheme R7. Anode reaction processes.

4) The cyclic voltammetry (CV) experiments for the nucleophiles should also provide.

Responses:

Thank you for the question. According to your suggestion, we have tested cyclic voltammetry (CV) of MeOH and pyrazole, as shown in Scheme R8. The result has been added to the revised manuscript and SI.

Scheme R8. The cyclic voltammetry (CV) of MeOH and pyrazole

5) How about other type of diazo compounds? Such as phenyldiazoacetate.

Responses:

Thank you for the suggestion. We have investigated the reaction by employing phenyldiazoacetate. However, the result indicated that most of phenyldiazoacetate

remained and a large of disulfides were observed by GC-MS (Figure R9).

Scheme R9. The experiment with ethyl 2-diazo-2-phenylacetate as substrate.

流路号:1 保留时间:18.615(扫描数:2524)
质量峰:352
原始模式:单个 18.615(2524) 基峰:127.05(8433117)
背景模式:无 组 1 - 事件 1 Scan

Figure R9. The detection of disulfides.

6) In the reaction with *N,N*-dimethylaniline, could this reagent be oxidized to the iminium ion species, followed by addition with diazo group?

Responses:

Thank you for the question. Firstly, we tested the anode potential of the reaction system. During the electrolysis, Scheme R10 showed that the anode potential was about 0.8 V from beginning to end.

Scheme R10. The anode potential of the reaction system.

Secondly, we have tested the oxidation potential of *N, N*-dimethylaniline, as shown in Scheme R11. The first oxidation peak (half peak potential, $E_{p/2}$) is about 0.8 V, which showed that *N, N*-dimethylaniline lost an electron and a proton to generate a carbon radical. The second oxidation peak (half peak potential, $E_{p/2}$) is about 1.4 V, which showed that the above carbon radical further lost an electron to generate the iminium ion species.

Scheme R11. The oxidation potential of *N, N*-dimethylaniline.

These results indicated the carbon radical was generated rather than the iminium ion species in electrolysis.

In addition, we have performed the constant-voltage experiment, as shown in Scheme R12. The target product could be isolated with 29% yield for 2h when the anode potential was set to 0.8 V.

Scheme R12. The constant-voltage experiment.

Finally, we have performed the experiment with pyrazole as a nucleophile, as shown in Scheme 13. The difunctionalization product of diazo was detected by GC-MS. However, the coupling product of *N,N*-dimethylaniline with pyrazole was not detected by GC-MS (Figure R10).

Scheme R13. The experiment with pyrazole as a nucleophile.

Figure R10. The detection of the difunctionalization product.

7) What happened on the Cathode in the reaction with “LiX”?

Responses:

Thank you for the question. The reaction was performed in undivided cell. When LiCl was used as nucleophile, the Cl⁻ diffused to anode area and formed the product. The proton was reduced to generate hydrogen at the cathode in electrolysis. For thiophenol system, the proton from thiophenol and H₂O got electron to generate hydrogen at the cathode. For aniline system, the proton from HOAc got electron to generate hydrogen at the cathode.

Scheme R14. The cathode reaction.

REVIEWERS' COMMENTS

Reviewer #1 (Remarks to the Author):

I thank the authors for the changes made. I think the manuscript is now ready for being published.

Reviewer #2 (Remarks to the Author):

The author has dealt with all the issues, and this paper can now be accepted now.

Reviewer #3 (Remarks to the Author):

Issues have been well addressed in the revision. In terms of novelty of the transformation, this reviewer recommends accepting the manuscript as it is.

Reviewer #1 (Remarks to the Author):

I thank the authors for the changes made. I think the manuscript is now ready for being published.

Response: Thanks to this *Reviewer* for his/her recognition of our work. We acknowledge again for the detailed comments and suggestions from the first reviewer.

Reviewer #2 (Remarks to the Author):

The author has dealt with all the issues, and this paper can now be accepted now.

Response: Thanks to this *Reviewer* for his/her recognition of our work. We acknowledge again for the detailed comments and suggestions from the second reviewer.

Reviewer #3 (Remarks to the Author):

Issues have been well addressed in the revision. In terms of novelty of the transformation, this reviewer recommends accepting the manuscript as it is.

Response: Thanks to this *Reviewer* for his/her recognition of our work.